# Experience and Prevalence of Dental Caries in Migrant and Nonmigrant Low-SES Families’ Children Aged 3 to 5 Years in Italy

**DOI:** 10.3390/children9091384

**Published:** 2022-09-14

**Authors:** Gianmaria Fabrizio Ferrazzano, Giulia Di Benedetto, Silvia Caruso, Giuseppe Di Fabio, Sara Caruso, Maria Elena De Felice, Roberto Gatto

**Affiliations:** 1UNESCO Chair in Health Education and Sustainable Development, Paediatric Dentistry Section, University of Naples “Federico II”, 80138 Naples, Italy; 2Department of Life, Health and Environmental Sciences, Paediatric Dentistry, University of L’Aquila, 67100 L’Aquila, Italy

**Keywords:** child, dental caries, dmft, low SES, migrant, Italy, Paediatric Dentistry

## Abstract

Dental caries is a public health problem in children and is more prevalent in low-socioeconomic-status groups. The aim of this study is to assess the association between migrant families and the prevalence of caries among young children in Italy. This is a cross-sectional study. In the age range of 3 to 5 years, a total of 266 migrant children and 301 nonmigrant children were examined in three Italian charity dental centers. All children had families with low SES. The dmft was determined by intraoral examination performed by six pediatric dentist specialists to assess their dental health. In this study, the prevalence of caries (71%) and the mean dmft (3.68; SD: 1.52) of migrant children were statistically significantly higher than the percentage (52%) and the mean dmft of the nonmigrant control group (3.10; SD: 1.65) with no differences between genders. For the migrant children, the mean (0.49; SD: 0.32) of restorations (filled teeth) was statistically significantly lower than that of nonmigrant children (1.20; SD: 0.48). This study highlights that dmft values and the prevalence of caries are higher in migrant children than in nonmigrant children. In addition, the control group shows a higher level of dental caries than the national mean.

## 1. Introduction

Dental caries is the most common chronic disease in both children and adults in the world. Severely decayed teeth can have a substantial negative impact on children’s health and well-being [1]. According to WHO data, 60–90% of school-age children around the world have caries. Additionally, in Italy, dental caries remains a significant public health problem, with a high prevalence. Indeed, at 4 years of age, 21.6% of children have caries; at 12 years, 43.1% are affected by caries, while in the age group between 19 and 25, 88.2% of people have carious lesions [2].

Since the presence of caries is a very common phenomenon in many patients treated with orthodontic appliances who have poor oral hygiene, a lot of innovative methods have been developed to reduce treatment times and caries as a consequence [3,4,5,6].

In the last few years, several programs of dental health improvement have been realized, having great success with the prevalence of dental caries, which has been significantly reduced [7], but this improvement seems not to involve the population of young children who are socially disadvantaged, such as migrants, in whom caries is still highly prevalent [8,9,10]. In fact, “migrant status” is still considered a social vulnerability for oral health [11,12].

In the history of industrialized countries (such as the USA), immigration began centuries ago and is radicated in the culture of those countries. In the last decades, the immigration phenomenon also in countries belonging to the European community has strongly increased, in particular due to job requests, education, escape from armed conflict, and poverty or climate change (European Commission, 2011) [13,14]. One of the problems among migrant people is food insecurity due to a sudden change in diet compared with one’s previous eating habits, which are linked to cultural traditions [15]. This factor, combined with a low socioeconomic setting and the high simple-sugar foods they consume in the host country, might increase a number of adverse health outcomes, mostly pertaining to children [16,17,18,19,20]. The lack of specific nutrients and of healthy diet, which are essential components for developing organisms, can cause micronutrient deprivation and induce illnesses, such as dental caries, anemia, and rickets. Indeed, economic and cultural factors could also worsen diet with overconsumption of low-cost, nutrient-poor products and beverages (junk foods), which are directly related to early childhood caries (ECC) [21,22,23,24]. The American Academy of Pediatric Dentistry defines ECC as “the presence of 1 or more decayed (non-cavitated or cavitated lesions), missing (due to caries), or filled tooth surfaces” in any primary tooth in a child 71 months of age or younger [25]. Lack of fluoride intake and healthy diet, poor oral hygiene, and low economic possibilities to access dental care predispose the development of ECC, which eventually may lead to pain, infection, and eating problems [26,27,28]. 

In Italy, only in the last 30 years, immigration from other countries, such as Asia, Africa, and East Europe, has undergone a notable acceleration, being a fairly recent phenomenon (since 2013, according to the Italian National Institute of Statistics (ISTAT)).

Results of a national survey on the oral health status of 4-year-old children, conducted by Campus et al. [29], showed significant associations between caries and nationality of the parents (non-Italian vs. Italian: *p* < 0.001). On the contrary, in another previous national survey of 12-year-olds by the same authors, there was no ethnicity-related increase in the prevalence of caries [30].

Another Italian study [31] analyzed the correlation between social vulnerability (low occupation level and migrant population) and the experience of tooth decay in a cohort of children aged 3–5 years in Veneto (region of Italy). It was observed that lower social class families’ children had more experience of tooth decay, and this prevalence in the immigrant group was significantly higher than in Italian children, and caries was found to be almost four times more severe in migrant ones.

Therefore, the aim of this study is to assess dental caries experience and provide the estimated prevalence in migrant and nonmigrant low-SES families’ children in Italy in order to develop oral health improvement policies for those particular socially vulnerable groups.

## 2. Materials and Methods

### 2.1. Data Source

This cross-sectional study was conducted from February 2020 to December 2021. The sample included children aged in the range of 3 to 5 years, with a total of 325 migrant children and 389 nonmigrant children. 

The exams were conducted in three charity dental centers (in Milan, north Italy; Rome, central Italy, and Naples, south Italy) supported by Caritas Italia and ‘In the Name of Life” Foundation between May and December 2021 by six pediatric dentist operators, three belonging to the Department of Life, Health, and Environmental Sciences Section of Pediatric Dentistry at the University of L’Aquila (Italy) and the three others belonging to the Unesco Chair in Health Education and Sustainable Development Pediatric Dentistry Section at the University of Naples Federico II (Italy). Visual and exploratory examinations were performed on dental chairs under artificial light using disposable oral trays containing dental mirrors, forceps, and explorers without X-ray assessment. All examiners were calibrated at the Unesco Chair in Health Education and Sustainable Development Pediatric Dentistry Section at the University of Naples Federico II (Italy), and the kappa test revealed a final score of k = 0.90 (CI 95% 0.795–0.952).

The inclusion criteria were: children aged 3–5 years and families’ low socioeconomic status. The exclusion criteria were: presence of severe general pathologies (heart disease, cancer, chronic respiratory diseases, mental diseases, musculoskeletal and gastrointestinal disorders, vision and hearing defects, and genetic diseases) and lack of informed consent. Considering these aspects, 266 migrant children and 301 nonmigrant children were included in the study. Based on exam data, the dmft index was determined for each child following the guideline from the World Health Organization, Geneva, Switzerland; that index refers to the total number of decayed (untreated caries) (d), extracted/missing due to caries (m) (as distinct from missing for reasons other than caries, such as natural exfoliation), and filled due to caries (f) teeth, for primary (lowercase) dentition.

### 2.2. Individual- and Family-Level Variables

The outcome variable for this study was the presence (vs. absence) of decay experience (i.e., dmft > 0), that is, whether a child had at least one primary tooth that was decayed, extracted/missing due to decay, or filled (vs. not). The threshold for the presence of dental decay experienced in this index is the detection of caries that has caused the enamel of the tooth to be cavitated. Demographic variables included age, gender, socioeconomic status, and country of origin (migrant vs. nonmigrant). Children were classified as “migrants” if the mother was from a “non-Western country” (Eastern Europe, Asia, Africa, Turkey, South, and Central America); the others were assigned the status of “nonmigrant”. The parents of all the participants completed a questionnaire to provide information and documentation about family sociodemographic background with regard to the annual income of the family. To certify the questionnaire, all the parents needed to bring ISEE certification, which is an official document that contains family income and patrimonial information necessary to describe the economic situation of a family in any given year.

### 2.3. Statistical Analysis

SPSS 22.0 (IBM; Armonk, NY, USA) was used for statistical data analysis. Measurement data were represented as mean and standard deviation. A Mann–Whitney U test was used to compare the dmft’s means, and a chi-squared test was used to compare the caries prevalence of the two samples. The significance level (α) was set to 0.05. Sampled children with missing values were excluded from analyses, and all analyses were based on the subset with complete data on all study variables.

## 3. Results

Among 567 children, 266 were migrant and 301 nonmigrant. Based on gender, 285 were male and 282 were female. All the families of the children included in the study had low SES, as stated in the parents’ questionnaire; therefore, that variable was not included in the analysis results.

Table 1 shows that the total percentage of caries in nonmigrant patients aged 3–5 years (52%) was statistically significantly lower (*p* < 0.05) than in the migrant group (71%). There was no statistically significant difference in the rates of caries between genders within the 3–5 age range of both groups. 

Table 2 presents the data obtained through intraoral examination and shows that the mean dmft index of migrant children (3.68; SD: 1.52) was statistically significantly higher (*p* < 0.05) than that of nonmigrant children (3.10; SD: 1.65) with no statistically significant difference between genders. Regarding the single values of dmft, the mean of the decayed teeth (2.36; SD: 1.26) of migrant children were statistically significantly higher than the mean (1.13; SD: 0.56) of those of nonmigrant children. The mean of the missing teeth was similar, with a mean value of 0.83 (SD: 0.25) for migrant children and 0.77 (SD: 0.15) for nonmigrant children. Moreover, for the migrant children, the mean (0.49; SD: 0.32) of restorations (filled teeth) was statistically significantly lower than those of nonmigrant children (1.20; SD: 0.48). All these values showed no statistically significant difference between genders.

## 4. Discussion

The aim of this study is to assess the experience and prevalence of dental caries between migrant and nonmigrant families and the prevalence of caries among young children in Italy.

In recent years, in industrialized countries, the prevalence of caries has significantly reduced, thanks to prevention. Despite this, the experience of caries in children remains a public health problem, especially in groups with a low socioeconomic level. Recent studies have shown that the relationship between dental caries and low socioeconomic level is well consolidated, while patients with a high socioeconomic level have better oral health [32]. Other studies have highlighted how the prevalence of caries increases in relation to a low SES, in both deciduous and permanent dentition [33,34]. The reasons for these differences, as already mentioned, are well known and are linked both to the difficulties in accessing health care and dental care for economic reasons and to the daily management of oral hygiene and prevention, as well as to incorrect eating habits. Furthermore, other studies have revealed that DMFT/dmft values, caries prevalence, and unmet restorative treatment needs index among migrant children were higher than those among nonmigrant ones [35].

In addition, the recent COVID-19 pandemic has been associated with several changes in the maintenance of children’s dental health: during the lockdown, it was found that children changed their daily habits. Some authors have pointed out an association between the increase in the frequency of taking snacks and sugary drinks, the decrease in the frequency of brushing teeth, and the postponement of oral health care to an increased risk of developing tooth decays [36,37].

During the first spread of the COVID-19 pandemic, most public and private territorial health facilities in most countries were closed; the consequences of the suspension of routine dental care and the delay in the management of oral hygiene have led an increase in dental emergencies, especially in the pediatric population [38], which was followed by higher access to first-aid facilities for dental emergencies, especially in the most disadvantaged categories of the population.

Considering these statements, the present study analyzed the oral health status in two groups of children (migrants and nonmigrants, both coming from low-SES families), and the extracted results showed that the difference in the mean dmft between the two groups was statistically significant (*p* < 0.05) and the oral health was worse in migrant children, confirming the relationship between social vulnerability and caries experience. In addition to this, the prevalence of caries and the dmft were consistently higher than the national mean even for the control group, and this is attributable to the fact that the control group is also part of a category of social vulnerability (low SES), which predisposes to a higher development of caries. Analyzing the components of the dmft index, the migrant children had more decayed (d) and less filled (f) teeth than the control group, thus denoting a greater need for dental care and absence of control of the carious pathology. Therefore, the values regarding filled teeth are statistically significantly higher for children belonging to the nonmigrant group, and this would mean, on the contrary, greater control of the carious pathology and greater access to dental care for this group. Conversely, the difference for missing teeth (m) was not statistically significant.

The results of this study provide a detailed insight into the oral health status of low-income families’ migrant and nonmigrant children. The results are in agreement with current epidemiological studies: the prevalence of caries and the dmft in the socially disadvantaged classes are higher than the national mean; in addition, migrant children compared with natives have a higher prevalence of caries and a greater need for dental care. Therefore, the present knowledge can be useful to design national epidemiological studies to evaluate the current state of caries in Italy to verify the impact of COVID-19 on children’s oral health, even in these high-risk groups featured in this study.

To be effective, these interventions should be easily accessible and engage both children and parents, such as implementing community/school health programs and giving information about preventive and curative aspects.

Another interesting intervention would be to involve women beginning in the gestational period so that they can be taught by several professionals, such as dentists, dental hygienists, and pediatricians, about oral health. It is really important that pregnant woman have information about oral hygiene techniques, about nutrition aimed at the prevention of caries in newborn children, and about the importance of the evaluation of a child’s oral cavity at 18–24 months by an oral health professional. All these strategies should be used in order to achieve the goal of zero caries in the pediatric population.

## 5. Conclusions

This study highlights that children aged 3 to 5 years with a low socioeconomic level show high values of tooth decay. Moreover, dmft values and the prevalence of caries are higher in migrant children than in nonmigrant children. Therefore, we can state that, in the lower-income classes, caries disease in primary dentition is still a public health problem today, especially in the migrant population. It is necessary to evaluate and monitor these situations in order to develop effective programs aimed at improving and, subsequently, maintaining, the state of oral health to intervene promptly and prevent the appearance of dental caries.

## Figures and Tables

**Table 1 children-09-01384-t001:** Prevalence of caries associated with gender in migrant and nonmigrant groups.

Children Aged 3–5 Years (567)	
Variables	Caries (%)
Migrant	
Female (145)	70.5 **
Male (121)	73.1
Tot. (266)	71
Nonmigrant	
Female (137)	54.6 **
Male (164)	49.2
Tot. (301)	52 *

* *p* < 0.05, the percentage of caries in nonmigrant children was statistically significantly lower than in migrant children; ** *p* > 0.05, there was no statistically significant difference in the percentage of caries between genders (chi-squared test).

**Table 2 children-09-01384-t002:** dmft index and distribution of data in migrant and nonmigrant groups.

Children Aged 3–5 Years (567)				
Variables	dt (SD)	mt (SD)	ft (SD)	dmft (SD)
Migrant				
Female (145)	2.34 (1.23)	0.79 (0.32)	0.48 (0.15)	3.61 (1.22) **
Male (121)	2.36 (1.18)	0.85 (0.33)	0.54 (0.22)	3.75 (1.44)
Tot. (266)	2.36 (1.26) *	0.83 (0.25)	0.49 (0.32) *	3.68 (1.52) *
Nonmigrant				
Female (137)	1.15 (0.52)	0.80 (0.32)	1.17 (0.45)	3.12 (1.15) **
Male (164)	1.12 (0.78)	0.73 (0.42)	1.25 (0.22)	3.05 (1.25)
Tot. (301)	1.13 (0.56)	0.77 (0.15)	1.20 (0.48)	3.10 (1.65)

* *p* < 0.05, the mean index of migrant children was statistically significantly different than that of nonmigrant children; ** *p* > 0.05, there was no statistically significant difference of the mean dmft index between genders (Mann–Whitney U test).

## Data Availability

The data of this study will be made available after a formal request to the authors.

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
