# Peer review of "Experience and Prevalence of Dental Caries in Migrant and Nonmigrant Low-SES Families’ Children Aged 3 to 5 Years in Italy"

_children, 2022, doi:10.3390/children9091384_

Round 1
Reviewer 1 Report
Manuscript title: Dental caries experience in migrant and not migrant low-SES families’ 2 children aged 3 to 5 years in Italy
The authors present results from a cross-sectional study to assess dental caries experience in migrant and not migrant low-SES families’ children in Italy, in order to develop oral health improvement policies for those particular socially vulnerable groups. In general, the manuscript is well organized. The studied population is considerable, and the statistic tools employed are appropriate to provide reliable data. The topic is not scientifically innovative; however, this paper provides epidemiological data that are crucial to public health planners. With the purpose of improve the quality of this paper, I do have some suggestions / comments / recommendations before being published.
Title
The authors could add the word “prevalence” to the title, since they also measured it: “Experience and prevalence of dental caries in migrant and not migrant low-SES families’ 2 children aged 3 to 5 years in Italy”
Since "experience" refers to the average caries index and "prevalence" to the percentage of subjects with caries.
Abstract
- As far as my information, the policy of MDPI is to have non-structured abstract.
- Please included the study design.
- Add a brief background of the research problem at the beginning of the abstract section.
- The authors use lowercase and uppercase letters to refer to the caries index. Given the age of the participants, they should refer to the caries index in lower case.
- The word "cohort" can confuse readers. It must be removed.
- The phrase "… and that would suggest a greater frequency of dental visits and treatments for the control group." should be removed from the results section. Since in the results section there should be no interpretation.
Introduction
- What is your hypothesis or research question?
Material and Methods
- Information about the sample size calculation and sampling method is not mentioned clearly. Was there any calculation of the sample size?
- What was the participation rate?
- How was socioeconomic status determined?
- Student's t-test is used when the data distribution is normal. In my experience conducting caries studies, the dmft index is never normally distributed. Therefore, they must use the Mann-Whitney test to compare the caries index between the migrant and non-migrant groups. If not, what is the normality test used by the authors to determine the normality of the variable dmft.
- The authors do not mention which test they used to compare caries prevalence between the migrant and non-migrant groups.
Results
- In tables, please add what statistical test the authors performed to report the p-value.
Discussion
- Please start by mentioning the objective of the study.
- Add a small paragraph about "future directions" for the current research problem.
Conclusion
- The last two paragraphs of the conclusion should be placed in the discussion section.
References
References must be properly cited.
Please read the instructions to the authors.
Author Response
Thanks author for the reviews sent, all the points you indicated have been corrected. Please see the attachment.
Best Regards.
Dr Di Benedetto Giulia

Reviewer 2 Report
Dear authors,
This is a simple but effective manuscript to demonstrate the impact of being from culturally and language diverse groups on oral health.
There are some comments that I have made on the paper. Some pertain to minor language issues while others are points to consider for greater clarification.
I hope that these help with improving your manuscript.

Author Response
Dear author,
thanks for the revisions sent, all changes have been made to the manuscript.
Please see the attachment
Best Regards
Dr Di Benedetto Giulia

Round 2
Reviewer 1 Report
Abstract
- As far as my information, the policy of MDPI is to have non-structured abstract. Delete the headers.
Statistical analysis section
The chi square test is for comparing percentages, not for comparing averages.
Author Response
Thanks author for the reviews sent, all the points you indicated have been corrected. The prevalences compared to the chi-square are expressed as a percentage.
Please see the attachment.
Best Regards.
Dr Di Benedetto Giulia
